# Electrochemical Mechanisms and Optimization System of Nitrate Removal from Groundwater by Polymetallic Nanoelectrodes

**DOI:** 10.3390/ijerph20031923

**Published:** 2023-01-20

**Authors:** Fang Liu, Zhili Zhang, Jindun Xu

**Affiliations:** Institute of Transportation, Inner Mongolia University, Hohhot 010070, China

**Keywords:** nanoelectrode, electrochemical workstation, reaction mechanism, electrochemical performance, optimization

## Abstract

Zn-Cu-TiO_2_ polymetallic nanoelectrodes were developed using Ti electrodes as the substrate. The reaction performance and pollutant removal mechanism of the electrodes were studied for different technological conditions by analyzing the electrochemical properties of the electrodes in the electrochemical system, using Ti, TiO_2_, Cu-TiO_2_, and Zn-Cu-TiO_2_ electrodes as cathodes and Pt as the anode. The Tafel curve was used for measuring the corrosion rate of the electrode. The Tafel curve resistance of the Zn-Cu-TiO_2_ polymetallic nanoelectrode was the smallest, so the Zn-Cu-TiO_2_ nanoelectrode was the least prone to corrosion. The electrode reaction parameters were determined using cyclic voltammetry (CV). Zn-Cu-TiO_2_ polymetallic nanoelectrodes have the lowest peak position and the highest electrochemical activity. The surface area of the electrode was determined by the time-current (CA) method, and it was found that the Zn-Cu-TiO_2_ polymetallic nanoelectrode had a larger surface area and the highest removal rate of nitrate. The Ti, TiO_2_, Cu-TiO_2_, and Zn-Cu-TiO_2_ electrodes also had higher removal rates for real groundwater, and the differences between the removal rates of nitrates for deionized water and real groundwater decreased as removal time increased. The Zn-Cu-TiO_2_ polymetallic nanoelectrode exhibited the highest removal rate for real groundwater. This study reveals the reaction mechanism of the cathode reduction of nitrate, which provides the basis for constructing electrochemical reactors and its application in treating nitrate-contaminated groundwater. A mathematical model of optimized working conditions was created by the response surface method, and optimum time, NaCl concentration, and current density were 93.39 min, 0.22 g/L, and 38.34 mA/cm^2^, respectively. Under these optimal conditions, the nitration removal rate and ammonium nitrogen generation in the process solution were 100% and 0.00 mg/L, respectively.

## 1. Introduction

Nitrate is a widespread pollutant in groundwater, and nitrate pollution has gradually developed into a major environmental pollution problem [1,2,3,4,5]. Nitrate contamination in groundwater is widely believed to have its source in human activity [6,7,8,9]. The following aspects contribute to nitrate pollution of groundwater: excessive use of chemical fertilizer, leaching and infiltration of solid waste, backflow irrigation of sewage, and excessive use of animal manure [8,9,10,11,12,13,14,15].

Since the 1960s, there have been reports of nitrate pollution in groundwater in the United States. In the following decades, Europe also reported nitrate pollution of groundwater [16,17]. The US Environmental Protection Agency conducted a groundwater quality survey in 1992, which investigated about 3 million people, including 43,500 infants, and determined that the nitrate content of drinking water exceeded the quality standard set by the World Health Organization (WHO) [18]. In the 1990s, Ermeydouil first mentioned that the concentration of nitrate in groundwater in the United States was 40–50 mg/L, and it frequently appeared [19]. In Los Angeles, the nitrate content in about 40% of well water in 1998 was higher than the standard level [8,12]. The Spalding study on groundwater quality conducted in some states of the United States also showed that in rural areas, the concentration of nitrate in more than 20% of wells exceeded the allowable limit for the drinking water standard [20]. Hiscock found that the concentration of nitrate pollutants in more than 100 wells in the U.K. exceeded the standard value. In Wales, for 125 sources of drinking water for 1.8 million people, the nitrate concentration was above the standard for groundwater [21]. Costa and Van Maanen reported that nitrate concentrations of 40–50 mg/L can be frequently observed in groundwater in France, Russia, and the Netherlands, and even up to 500–700 mg/L in some wells [22,23]. 

Nitrate contamination of groundwater needs to be taken very seriously by our people. Excessive consumption of groundwater with high levels of nitrate ions can have adverse effects on human health and, in more serious cases, can lead to cancer and other diseases. Therefore, effective methods should be developed to treat nitrate in groundwater to meet the requirements of drinking water quality standards. There are many ways to deal with nitrate problems, but electrochemistry is widely used by experimentalists because of its simple operation equipment, simple methods, easy control of experiments, absence of excess impurities, and low investment costs. However, the mechanism of electrochemical reaction is still unclear, its nitrate removal efficiency is low, and the electrode surface activity is poor, which limits the practical application of electrochemical methods.

The most critical issue in the use of electrochemical nitrate removal technology is the need to strictly control the removal of nitrate by electrode materials and energy consumption [24,25]. The microstructure of the electrode material is a central factor in controlling the removal efficiency of the electrode. Therefore, the electrochemical method can be improved by selecting appropriate materials for electrode development and studying reaction mechanisms [26,27]. Titanium is often used as an electrode because of its good corrosion resistance, high conductivity, high strength, nontoxicity, harmlessness, and economic advantages. The removal rate of nitrate using a single metal Ti electrode was only 21.3%, while the removal rate using a TiO_2_ nanoelectrode was significantly multiplied beyond 51.2%, which is a significant but still an unsatisfactory increase. Therefore, we believe that we can develop a modified nanoelectrode to study the reaction mechanism and further improve the nitrate removal rate.

The purpose of this study is to study the mechanism of nitrate removal using a new type of polymetallic nanoelectrode. The study fully demonstrates the practical application of environmentally friendly electrochemical technology and provides a practical case for the subsequent research and development of electrochemical technology and effectively solving the problem of nitrate pollution in groundwater. From the perspective of ensuring the safety of drinking water, the conclusions of this study are very important.

## 2. Materials and Methods

### 2.1. Construction of the Electrochemical System 

In the electrolytic cell, pour 100 mL acetic acid at 1:10 and add 0.05 wt% HF. Graphite is used as cathode, titanium plate is used as anode, and TiO_2_ nanoelectrode is prepared. Pour 100 mL of CuSO_4_·5H_2_O at a concentration of 180 g/L into the electrolyzer and add 3.270 mL of pure H_2_SO_4_ analysis to adjust the pH. Cu-TiO_2_ polymetallic nanoelectrode was made by using TiO_2_ nanoelectrode as cathode and Cu sheet as anode, with a current of 0.15 A and a power-on time of 10 s. Pour 100 mL of 160 g/L KCl and 80 g/L ZnCl_2_ into the electrolytic cell and add 0.5 mL of HCl to adjust pH. The Cu-TiO_2_ double-layer nanoelectrode was used as the cathode, the Zn plate was used as the anode, the current was adjusted to 0.25 A, and the Zn-Cu-TiO_2_ polymetallic nanoelectrode was fabricated after energizing for 10 s. The schematic diagram of Zn-CuTiO_2_ polymetallic-nanoelectrode-making process as Figure 1.

### 2.2. Analytical Methods and Calculations

The microstructures of the Zn-Cu-TiO_2_ polymetallic nanoelectrode were analyzed via scanning electron microscopy (SEM, Zeiss Merlin JEM-6301, Carl Zeiss AG, Oberkochen, Germany) The nitrate and ammonia concentrations were determined via molecular absorption spectrophotometry using an ultraviolet spectrophotometer (DR6000, HACH, Danaher Company, Loveland, CO, USA). The ammonia concentrations of the treated solutions were determined via Nessler’s reagent spectrophotometry. The nitrate removal efficiency was calculated using Equation (1)
(1)$$Q=\frac{C_{0}-C_{1}}{C_{0}}\times 100\%$$

*C*_0_—initial nitrate concentration, mg/L.*C*_1_—final nitrate concentration after the reaction, mg/L.

### 2.3. Experiment of Electrochemical Workstation

In this study, the traditional three-electrode method was used. The working electrodes were a pure Ti plate, the TiO_2_ nanoelectrode, the Cu-TiO_2_ bimetallic nanoelectrode, and the Zn-Cu-TiO_2_ polymetallic nanoelectrode, respectively. The effective working area of all working electrodes was 1 cm^2^. The paired electrode material for the construction of this experiment was platinum and the reference electrode was available as a saturated calomel electrode. The experimental electrodes were placed in a glass electrolyzer and tested at room temperature. Ultra-pure water was added to the ultra-pure reagent to prepare the solution and the resistivity of the solution was tested and measured to be 18.2 MΩ⋅cm. In the study, the Tafel curves were acquired, and in order to explore in depth the electrochemical performance of the cathode in electrochemical reactions, cyclic voltammetry and chronoamperometry were used to carry out studies of the target parameters. The instrument used was the Shanghai Chen Hua CHI660D electrochemical workstation. The curves were recorded using the ALS software (ALS Limited, Model 660D, Fusida Electronic Technology (Suzhou) Co., Ltd., SuZhou, China). The analysis was carried out by integrating the obtained parameter data using OriginPro software.

## 3. Results and Discussion

### 3.1. Comparison of Electrode Removal Rate

Figure 2 compares the nitrate removal efficiencies for different electrodes. When the conditions of a current density of 30 mA/cm^2^ and electrolysis time of 90 min were satisfied, the removal rates of nitrate from water by single metal materials Ti, TiO_2_ nanoelectrode, Cu-TiO_2_ bimetallic nanoelectrode and Zn-Cu-TiO_2_ polymetallic nanoelectrode were 21.3%, 51.2%, 65.1% and 97.5%, respectively. The Zn-Cu-TiO_2_ polymetallic nanoelectrode exhibited the highest nitrate removal rate.

### 3.2. Characterization of the Zn-Cu-TiO_2_ Polymetallic Nanoelectrode

The surface morphology and elemental composition of the Zn-Cu-TiO_2_ polymetallic nanoelectrodes were analyzed and processed using scanning electron microscopy (SEM) and energy dispersive spectroscopy (EDS) techniques, respectively, in turn. The bottom hole is the TiO_2_ base layer, the gray irregular particles in the interlayer are a copper coating layer, and the top white irregular grain layer is a galvanized layer. From Figure 3a, it can be seen that there are many small particles of constituent substances on the TiO_2_ nanoporous matrix in the Zn-Cu-TiO_2_ polymetallic nanoelectrodes. Energy dispersive spectra of the Zn-Cu-TiO_2_ polymetallic nanoelectrode are shown in Figure 3b. Copper and zinc are uniformly distributed on the porous titanium dioxide nanosubstrate. Figure 3c shows the distributions for different elements, which further proves that copper and zinc are uniformly distributed.

### 3.3. Mechanism of Electrochemical Reaction

Electrochemical workstations were set up to measure ion content using electrochemical methods, which are commonly used in electrochemical research. The advantages of using the electrochemical workstation are as follows: testing is simple, sensitivity is high, and the practicability is satisfactory. The electrochemical workstation can be used to perform cyclic voltammetry, AC voltammetry, AC impedance, potentiometric titration, and amperometric titration. The system can operate in two-electrode or three-electrode operation. When operating in two-electrode mode, there is no reference electrode; When operating in three-electrode mode, the system consists of a working electrode, a counter electrode, and a reference electrode. There is only one more sensitive electrode in the four-electrode mode compared to the three-electrode mode of operation, which is used to measure electrochemical parameters at the liquid/liquid interface. Among these modes, the three-electrode mode is the most commonly used, because this mode can meet accuracy requirements and is relatively simple. Nitrate is first reduced to nitrite after obtaining electrons at the cathode, and then nitrite continues to obtain electrons that are reduced to produce nitrogen, nitric oxide, etc. The reaction diffusion process of nitrate at the cathode is as follows:NO_3_^−^ + H_2_O + 2e^−^ = NO_2_^−^ + 2OH^−^
NO_3_^−^ + 3H_2_O + 5e^−^ = (1/2)N_2_ + 6OH^−^
NO_2_^−^ + 5H_2_O + 6e^−^ = NH_3_ + 7OH^−^
NO_2_^−^ + 4H_2_O + 4e^−^ = NH_2_OH + 5OH^−^
2NO_2_^−^ + 4H_2_O + 6e^−^ = N_2_ + 8OH^−^
2NO_2_^−^ + 3H_2_O + 4e^−^ = N_2_O + 6OH^−^
NO_2_^−^ + H_2_O + e^−^ = NO + 2OH^−^
N_2_O + 5H_2_O + 4e^−^ = 2NH_2_OH + 4OH^−^
2H_2_O + 2e^−^ = H_2_ +2OH^−^ (side reaction)

#### 3.3.1. Measurement of Metal Corrosion Rate Using the Tafel Curve Method

The electrochemical methods for measuring the corrosion velocity of metallic materials include the Tafel curve method, three-point method, constant-current transient method, and AC impedance method. The Tafel curve refers to a curve that conforms to the Tafel relationship, and generally refers to the section of the strong polarization region in the polarization curve. The Tafel curve method is an electrochemical method based on the electrochemical nature of metal corrosion. In the present study, the parameters of each electrode material were determined. To guarantee the accuracy of the experimental results, the side of the electrode plate plastic sealed with epoxy resin was put in a 100 mg/L NaNO_3_ solution. The scanning rate was 0.005 V/s, the voltage range was −0.4~0.4 V, and scanning was performed three times. The results are listed in Table 1. This method uses Tafel curves to characterize polarization. The logarithm of the longitudinal coordinates at the intersection point of two polarization curves is the corrosion current, while the abscissa is the corrosion voltage

The corrosion of the electrode is usually determined according to the corrosion potential and corrosion current density. Where the corrosion current density indicates the current state of corrosion in the cell, the current density is usually calculated by dividing the corrosion of the electrode by the test area; however, since the effective areas of the several working electrodes investigated in the present study were the same (1 cm^2^), the corrosion current was directly used for corrosion quantification. The corrosion potential of the electrode is determined, and if the data are characterized as positive, then the higher the corrosion current density, the stronger the electrochemical activity in the electrode. The corrosion potential and corrosion current of the Zn-Cu-TiO_2_ polymetallic nanoelectrode were 0.011 V and 0.9629 × 10^−7^ A, respectively, and it is obvious from Table 1 that these two parameters are significantly higher than those of the other electrodes. According to Figure 4, the Ti electrode, Ti nanoelectrode, and Cu-TiO_2_ bimetallic nanoelectrode have clear cathode polarization areas and anode polarization areas, while the Zn-Cu-TiO_2_ polymetallic nanoelectrode only has the cathode polarization area in the −0.109 V~−0.049 V range, without a clear anode polarization area. The Zn-Cu-TiO_2_ polymetallic nanoelectrode exhibits the most positive corrosion potential, the most corrosive current, and the least resistance in the Tafel curve. This suggests that the Zn-Cu-TiO_2_ polymetallic nanoelectrode is not prone to corrosion and is relatively efficient for the removal of pollutants. Therefore, when used for nitrate reduction, the removal rate is the highest, which is consistent with previous experimental results.

#### 3.3.2. Cyclic Voltammetry for Measurement of Electrode Reaction Parameters

Cyclic voltammetry is one of the most important electrochemical analysis methods, and it is often used to test the electrochemical properties of nanoscale electrodes. The cyclic voltammetry curves are a common method of determining the relationship. The principle is to add a cyclic voltage between the reference electrode and the working electrode, and then record the current obtained at the working electrode in relation to the applied voltage to obtain a valid linear relationship. The positive-sweep anodic oxidation process is basically a continuous process from negative to positive voltage scanning, corresponding to the peak of the oxidation peak. The reverse process is the negative sweep cathodic reduction process, corresponding to the peak value of the reduction peak.

Cycle voltammetry test results are shown in Figure 5. To eliminate the possible effects of other materials, blank experiments were performed using 0.5 g/L of Na_2_SO_4_, with 0.5 g/L Na_2_SO_4_ was added to 100 mg/L of NaNO_3_ solution as the working electrode in the experimental group, using a voltage range of 1.2–0.2 V and a scan rate of 0.02 V/s. Due to the reduction effect of different electrodes on nitrate, the reduction peak value in the cyclic voltammetry curves was of concern in this study. As curve A in Figure 4 shows, in the blank experiment with only sodium sulfate, the scanning curve is relatively flat and the reduction peak is barely resolvable, indicating that the four electrodes had no effect on sulfate in water. After adding sodium nitrate, clear reduction peaks appear, as shown in curve B in Figure 5.

According to Figure 5, reduction peaks for the Ti electrode, Ti nanoelectrode, Cu-TiO_2_ bimetallic nanoelectrode, and Zn-Cu-TiO_2_ polymetallic nanoelectrode are observed at −0.9 V, −0.7 V, −0.65 V, and −0.35 V, respectively, and the peaks represent the NO_3_^−^ reduction to NO_2_^−^, which is consistent with some previous reports. The lower the reduction potential, the more likely the pollutant is to react for removal. By comparison, the current fluctuation ranges of the Ti nanoelectrode, Cu-TiO_2_ bimetallic nanoelectrode, and Zn-Cu-TiO_2_ polymetallic nanoelectrode are larger than that of the Ti electrode, and the peak of the latter is located at a lower potential, indicating that its electrochemical activity is higher than that of the Ti electrode plate. However, the Zn-Cu-TiO_2_ polymetallic nanoelectrode exhibits the lowest peak voltage, which is easier to reach in electrochemical experiments, and exhibits obvious ability to reduce nitrate. Therefore, the structure of the double electrode layer of the nanoelectrode prepared under this condition is more conducive to the reduction of nitrate. The main reason is that the double electrode layer structure enables more nitrate ions to enter the double electrode layer and enhances the electrochemical characteristics of the electrode. 

#### 3.3.3. Measurement of the Actual Surface Area of the Electrode Using the Time Measurement Method

The electrodes in this study were nanoelectrodes, which have many holes and can have high porosity and specific surface area. Therefore, for the same nominal surface area, the true surface area may vary greatly across different electrodes, which in turns leads to variations in the electrochemical performances of the electrodes. Chronoamperometry can be used as a method to determine the electrochemical activity and the actual surface area of the electrode. In this study, the constant potential square wave method was adopted. The method is used to further measure the true surface area of the electrode more effectively by measuring the double-layer capacitance in the electrode; and to make a more targeted comparative analysis by using the double-layer capacitance in time, so that the true surface area of the electrode meets the calculation requirements. According to the double-layer theory introduced by previous generations, when the electrode is immersed in the electrolyte, a double-layer confined space is formed between the electrode and the contact surface of the solution. At present, the absolute value of the electric double layer cannot be measured, but if the electrode is charged or discharged with a slightly varying potential, the microvariable, ∆q, of the corresponding charge amount can be measured, so the differential capacitance value of the electric double layer of the electrode can be obtained. Thus, we consider the electrode electric double-layer capacitance, C_d_, because pure mercury is distributed on a smooth surface. The actual surface area can be derived from the apparent area of pure mercury being observed and measured. The double-layer capacitance value of the mercury electrode is 20 μF/cm^2^ (used as a standard value) as CN, which can represent the capacitance value present per unit of actual area. The measured double-layer capacitance value, C_d_, is divided by C_N_ to calculate the total true surface area of the electrode.

An amount of 100 mg/L of NaNO_3_ was added to the electrode of 0.5 g/L of Na_2_SO_4_ solution and the timing current was tested at a constant voltage of −0.7 V. The results can be understood in detail in Figure 6. It is obvious from the figure that the cathodic current density for nitrate reduction in the initial state is 3.4 mA/cm^2^ and then will drop rapidly to a stable value.

In addition, in the steady state, the current density values of the metal nanoelectrodes are all much higher than those of the single metal material electrodes, and the Zn-Cu-TiO_2_ multimetal nanoelectrode has the highest current density among them. Therefore, it can be found that the Zn-Cu-TiO_2_ polymetallic nanoelectrode has the best electrochemical activity and can effectively remove nitrate ions to a certain extent, which is approximately the same as the previous cyclic voltammetry test results.

The actual surface areas in the electrodes of each material were calculated to be 17.0, 80.7, 132.9, and 159.3 cm^2^, respectively. The real surface areas of the TiO_2_ nanoelectrode, Cu-TiO_2_ bimetallic nanoelectrode, and Zn-Cu-TiO_2_ polymetallic nanoelectrode were three to eight times larger than that of the Ti electrode plate. It is obvious from the data that the Zn-Cu-TiO_2_ polymetallic nanoelectrode has a large actual surface area, and this electrode will be in full contact with nitrate and remove it effectively during the reaction; thus, more nitrate molecules are removed from the electrode surface, significantly improving the removal rate of nitrate.

### 3.4. Optimization of Nitrate Removal Effect

#### 3.4.1. Design of Experiments

The response surface model was developed, in which the dependent variables were set as ammonia concentration and nitrate concentration; the three independent variables were current density, electrolysis time, and sodium chloride dosage; and each variable was set at high, medium, and low levels of +1, 0, and −1, depending on the magnitude. the experiments were conducted exactly in the designed order to avoid external interference. The concentrations of nitrate and ammonia nitrogen obtained under different reaction conditions are shown in Table 2.The repeated use of the center point was used to perform a valid error assessment and to achieve reliability validation. The significance index of this experimental model was determined based on the *p*-value. Ruthenium dioxide material and copper-zinc-titanium material were used as electrodes for the anode and cathode, respectively.

#### 3.4.2. Reaction Equation

Under the premise of keeping the other parameters unchanged, only one parameter was changed to study the effects of different parameters on the nitrate removal rate and ammonia nitrogen yield. The expected results of optimization are the highest nitrate (Y_1_) removal and the lowest ammonia nitrogen (Y_2_) yield. After calculation, the relationship between the reaction function and independent variables can be obtained as Equations (2) and (3):Sqrt(Nitrate removal rate) = 0.22365 + 0.15546A + 0.11808B + 1.12051C − 4.21926 × 10^−4^AB − 0.044918AC + 3.98797 × 10^−3^BC − 1.02582 × 10^−3^A^2^ − 4.75769 × 10^−4^B^2^ + 0.13329C^2^(R^2^ = 0.9951)(2)
Sqrt(Ammonia production) = 1.23662 − 0.017476A − 1.65783 × 10^−3^B − 4.72033C − 8.63322 × 10^−5^AB − 0.013716AC − 4.99009 × 10^−3^BC + 3.83342 × 10^−4^A^2^ + 4.02081 × 10^−5^B^2^ + 7.42363C^2^(R^2^ = 0.976)(3)

#### 3.4.3. Analysis of the Response Models

According to the test results, finding out the factors that have a significant effect and finding out at what level and process conditions can make the index optimal to achieve the purpose of high quality and high yield, are the problems that the analysis of variance (ANOVA) solves. The results of the ANOVA are shown in Table 3 and Table 4. If the confidence interval of the response function is up to 95%, then its prediction is, to some extent, representative of the actual result (R^2^ > 0.95). If no variables exist in the analysis, it will cause a serious bias in the results, so variables need to be established in the ANOVA. In addition, the ANOVA usually compares the model variables with the rest of the variables by way of an F-test. The closer the variables are, the closer the ratio is to one. The *p*-value is used to test whether the solution hypothesis holds. If the null hypothesis is true, *p* < 0.05 means that the influence is significant. Lack of fit >0.05 indicated that the misfit did not show that the regression equation was well fitted.

As can be seen from Table 3, except for C, the influence of other relational parameters was significant. There was a linear relationship between the current density parameter A (*p* < 0.0001) and the reaction time parameter B (*p* < 0.0001), while the linear relationship between the NaCl content parameter C (*p* = 0.5456) and the two was not obvious. In other words, the removal efficiency of nitrate was directly related to current density and reaction time, but not directly related to NaCl content. The ANOVA of the response surface quadratic model shows that F = 1.19, R^2^ = 0.9951, and probability < 0.0001, indicating that the model is highly relevant and that the experiment is reliable and accurate. Lack of fit = 0.42, indicating that the misfit was not significant, and the established regression equation fitted well.

Because the p-value is less than 0.05, it means that this parameter has a significant effect on the reaction. It can be seen from Table 4 that the effects of current density (*p* = 0.042) and sodium chloride addition (*p* < 0.0001) on the reaction are significant, while the reaction time parameters (*p* = 0.5161) have no significant effect on the reaction. Therefore, the current density and the small changes will affect the NaCl content of ammonia nitrogen production, but time will not affect the generation of ammonia nitrogen. Based on the ANOVA of the response surface quadratic model with parameters F = 5.17, R^2^ = 0.976 and probability < 0.0001, it can be inferred that the model is strongly linked, and the experimental results are representative. Lack of fit = 0.07, indicating that the misfit was not significant, and the established regression equation fitted well.

The experimental objective of this study was to achieve a faster peak efficiency of nitrate ion removal and to reduce ammonia nitrogen production to some extent effectively. Based on this purpose, the current density, reaction time, and NaCl content of this experiment were tended to be set to a minimum value of about 0.1, while the nitrate ion removal rate and ammonia nitrogen production were tended to be set to a maximum value of 1.0. The lowest value of nitrate removal rate was set as 80%, the highest value was set as 100%, and the lowest value of ammonia nitrogen production was set as 0. The response surface curve can be clearly seen in Figure 7. Figure 8 fully demonstrates the expectation of each item and differs from the overall expectation. The overall expectation value was D = 0.901.

In order to verify the effectiveness of process parameter optimization, repeated experiments were carried out on the desired data results obtained by optimization. Nitrate was removed by electrolysis under a current density of 38.34 mA/cm^2^, a reaction time of 93.39 min, and a NaCl content of 0.22 mg/L. After 93.39 min of electrolysis, the removal rate of nitrate was predicted to be 100%, and the production of ammonia nitrogen was 0.02 mg/L, while the actual experimental results showed that the removal rate of nitrate was 99.71%, and the production of ammonia nitrogen was 0.00 mg/L. Therefore, the prediction is basically consistent with the actual experimental results, which indicates that box–behnken design and the expectation equation can be effectively used to optimize the electrochemical method for removal of nitrate and ammonia nitrogen production.

### 3.5. Removal of Nitrate from Real Groundwater

The results presented above suggest that the developed Zn-Cu-TiO_2_ polymetallic nanoelectrode has good removal efficiency with respect to nitrate; nevertheless, the ultimate goal of this project was to determine whether the developed methods can be applied in realistic scenarios. To validate the efficiency of the developed electrode with respect to nitrate removal from real groundwater, real groundwater should be used instead of deionized water for experimentation. The nitrate content in the selected groundwater in daily life is often not obvious in the test phase of the phenomenon, which will cause a large error in the experiment; therefore, the addition of 50 mg/L of nitrate in real groundwater can effectively improve the achievement of the purpose. From Figure 9, the nitrate removal rates in the two types of water at different electrodes with different electrolysis times can be seen. It is clearly known from the experimental results that Zn-Cu-TiO_2_ polymetallic nanoelectrodes, after 90 min of electrolysis, have a significant removal effect, and the removal rate of nitrate in the two types of water can be as high as 87.9% and 95.1%. After 120 min of electrolysis time, the nitrate removal rates of the two waters were highly similar, up to 95.3% and 96.3%. After 180 min of electrolysis time, the nitrate removal rate in both waters can reach an amazing 99.0% and 100.0% (this may be because the remaining amount of nitrate in the solution after the end of the reaction is below the detection limit), which is extremely efficient. According to the above experimental results, it was found that the new Zn-Cu-TiO_2_ polymetallic nanoelectrode constructed in this study shows a strong ability to remove nitrate from real groundwater, which can reach an amazing 100%, and the difference in nitrate removal rate between the two waters gradually becomes smaller with the gradual increase in electrolysis time.

## 4. Conclusions

In this study, the Ti electrode, Ti nanoelectrode, Cu-TiO_2_ bimetallic nanoelectrode, and Zn-Cu-TiO_2_ polymetallic nanoelectrode were used as the cathodes, while Pt electrode material acted as the anode. By characterizing the differences in electrochemical properties that exist between different electrodes in an electrochemical system, the reaction performance and pollutant removal mechanisms of the reactor under different conditions are explored. The metal corrosion rate, electrode reaction parameters, and actual surface areas of the electrodes were studied using the Tafel curve, cyclic voltammetry, chronometric current, and other electrochemical methods. The removal effects of the four electrodes were also compared and studied for real groundwater. The following conclusions were made:(1)Tafel curves, which are used to measure the corrosion rate of metal, indicated that the Zn-Cu-TiO_2_ polymetallic nanoelectrode exhibited the most positive corrosion potential, largest corrosion current, and least resistance; thus, this electrode is concluded to be the least vulnerable to corrosion.(2)Cyclic voltammetry, which was used for determining the electrode reaction parameters, revealed that the peak position for the Zn-Cu-TiO_2_ polymetallic nanoelectrode was the highest, and its electrochemical activity was the highest.(3)By measuring the Zn-Cu-TiO_2_ polymetallic nanoelectrode using chronoamperometry, it was found that the true surface area was larger than the original surface area and the electrochemical activity of the electrode was higher, which led to a greater increase in nitrate removal.(4)The removal rate of nitrates from real groundwater by the electrodes had the same effect as that of laboratory-deionized water. With increasing electrolysis duration, the difference between the nitrate removal rate in actual groundwater and the nitrate removal rate in deionized water becomes smaller and smaller. Among the tested electrodes, the Zn-Cu-TiO_2_ polymetallic nanoelectrode demonstrated the best removal of nitrates from real groundwater.

## Figures and Tables

**Figure 1 ijerph-20-01923-f001:**
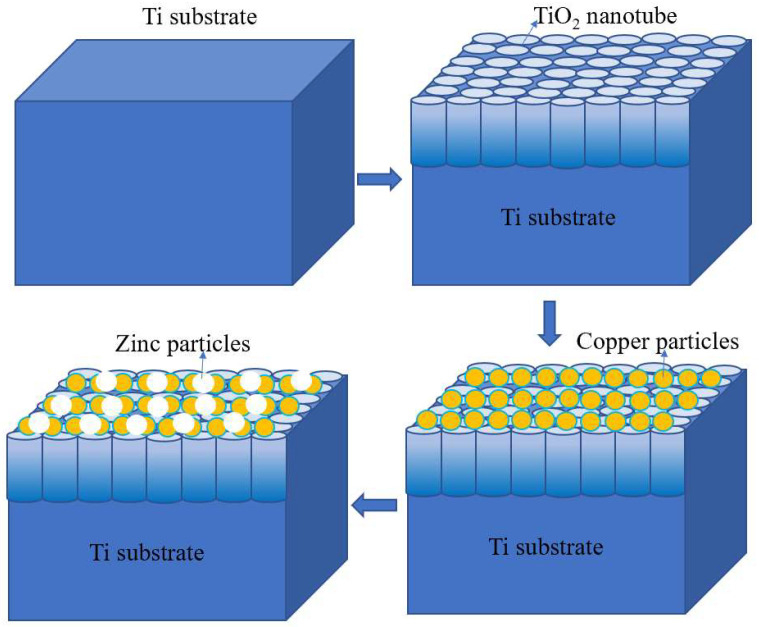
Schematic diagram of Zn-Cu-TiO_2_ polymetallic-nanoelectrode-making process.

**Figure 2 ijerph-20-01923-f002:**
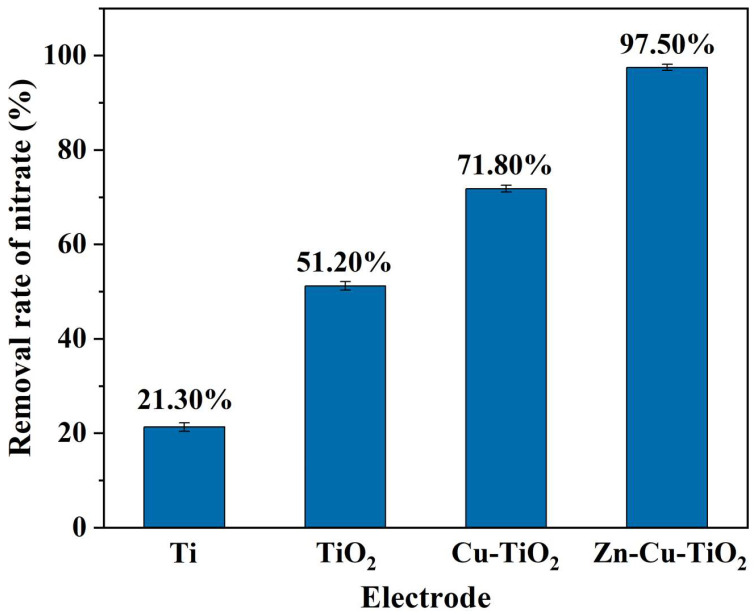
Electrode removal rate for each material.

**Figure 3 ijerph-20-01923-f003:**
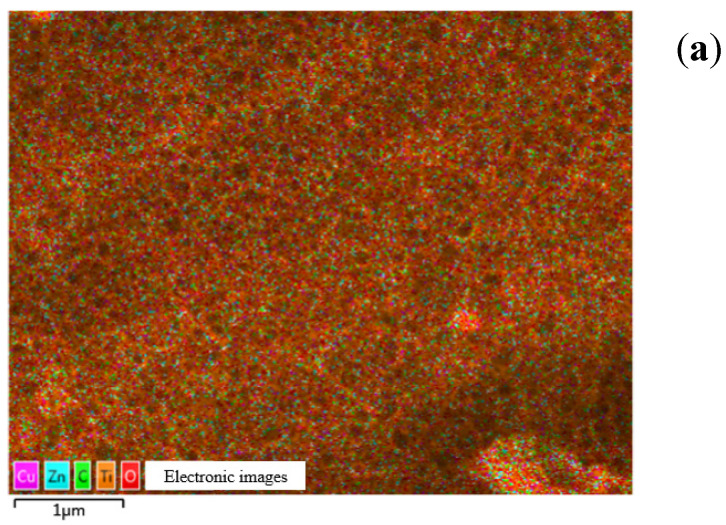
Characterization of the Zn-Cu-TiO_2_ polymetallic nanoelectrode in terms of its (**a**) surface morphology, (**b**) distribution of elements, and (**c**) distributions of individual elements.

**Figure 4 ijerph-20-01923-f004:**
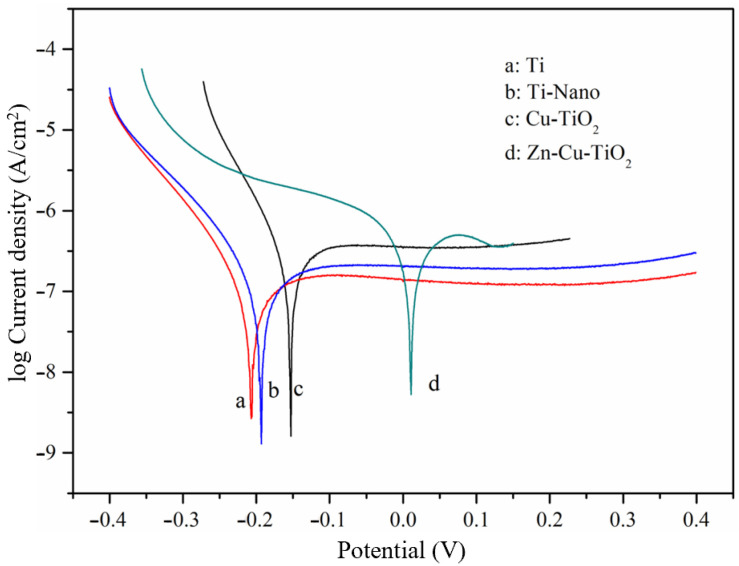
Tafel polarization curves for the different electrodes.

**Figure 5 ijerph-20-01923-f005:**
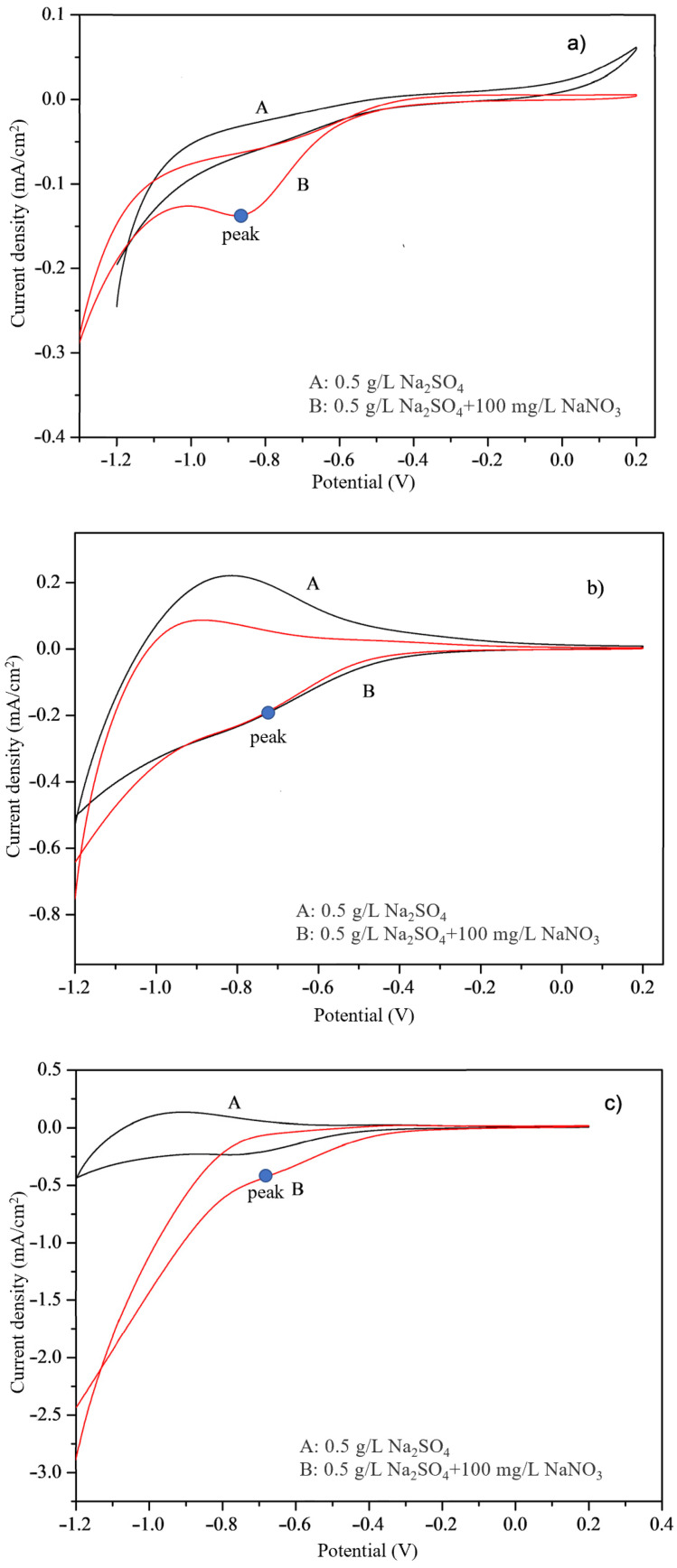
Cyclic voltammograms for (**a**) Ti, (**b**) Ti-nano, (**c**) Cu-TiO_2_, and (**d**) Zn-Cu-TiO_2_ electrodes.

**Figure 6 ijerph-20-01923-f006:**
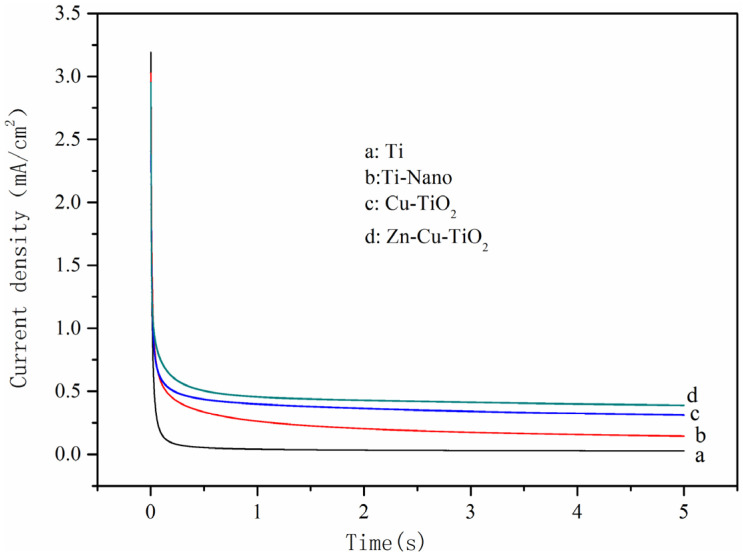
Current density vs. time for the different electrodes.

**Figure 7 ijerph-20-01923-f007:**
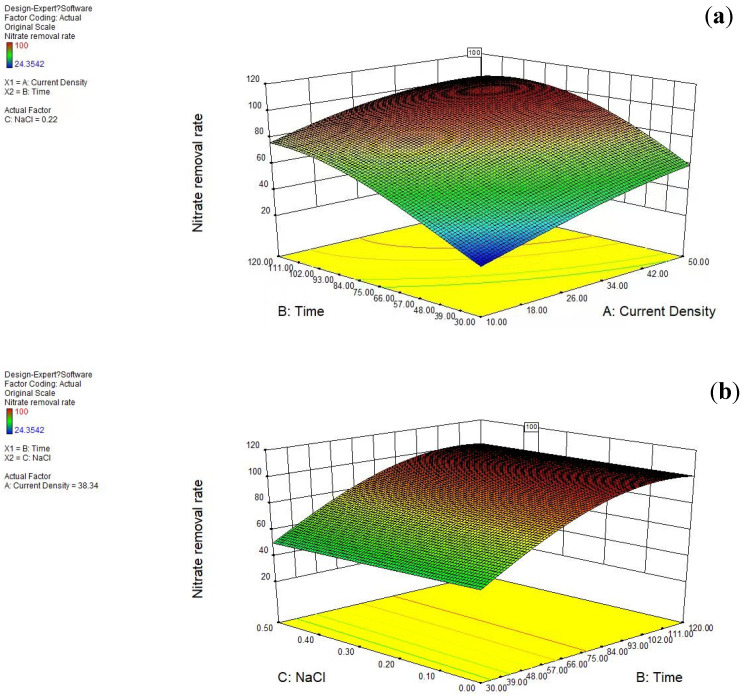
(**a**) Current density and reaction time, (**b**) reaction time and NaCl content, and (**c**) NaCl content and current density on nitrate removal of 3D three-dimensional standard deviation.

**Figure 8 ijerph-20-01923-f008:**
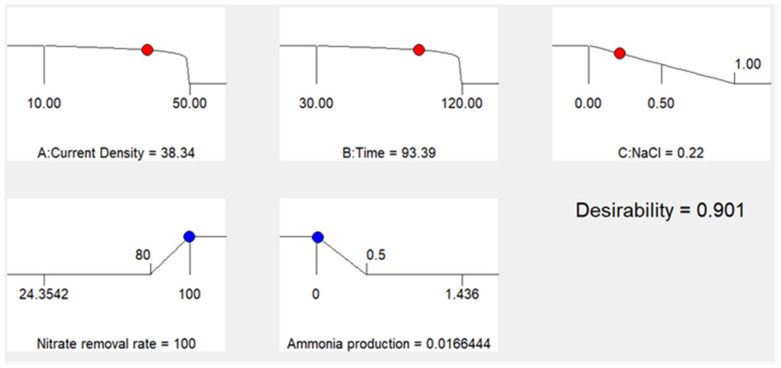
Optimization of target variables.

**Figure 9 ijerph-20-01923-f009:**
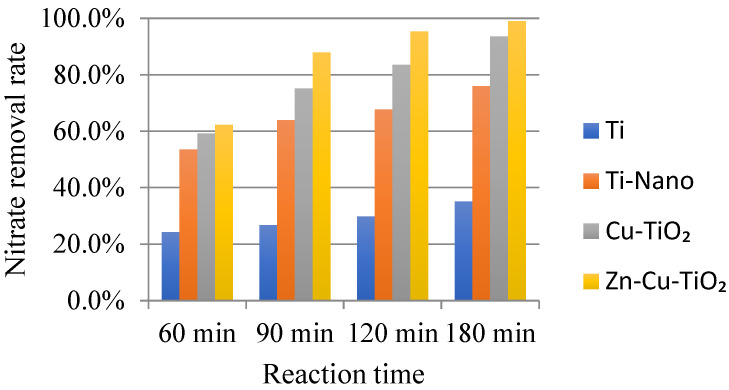
Comparison of the removal efficiencies of nitrate from deionized water and real groundwater for the different electrodes.

**Table 1 ijerph-20-01923-t001:** Results of the metal corrosion rate test.

Electrode	Corrosion Voltage (V)	Corrosion Current (A)	Tafel Curve Resistance (Ω)
Ti	−0.207	1.479 × 10^−7^	236,347.8
Ti-Nano	−0.193	1.975 × 10^−7^	192,950.7
Cu-TiO_2_	−0.153	3.131 × 10^−7^	74,590.2
Zn-Cu-TiO_2_	0.011	9.629 × 10^−7^	67,327.2

**Table 2 ijerph-20-01923-t002:** Observed nitrate and ammonia concentrations.

Run	Factor 1	Factor 2	Factor 3	Response 1	Response 2
	A: Current Density	B: Time	C: NaCl	Nitrate Removal Rate	Ammonia Production
	mA/cm^2^	Min	g/L	%	g/L
1	30	75	0.25	87.64	0.017
2	30	75	0.25	88.05	0.001
3	10	120	0.25	75.03	0.235
4	30	120	0.5	100.00	0.023
5	10	30	0.25	24.35	0.15
6	30	75	0.25	82.29	0
7	50	75	0.5	92.90	0.021
8	50	120	0.25	100.00	0.001
9	30	75	0.25	82.47	0
10	30	30	0.5	43.14	0.012
11	50	30	0.25	60.71	0.06
12	30	30	0	43.84	0.868
13	10	75	0.5	65.65	0.112
14	30	120	0	97.50	1.536
15	50	75	0	99.70	1.304
16	30	75	0.25	84.76	0
17	10	75	0	57.01	1.118

**Table 3 ijerph-20-01923-t003:** ANOVA analysis (Y_1_).

Source	Sum of Squares	df	Mean Square	F Value	*p*-Value Prob > F	
Model	33.86	9	3.76	157.71	<0.0001	significant
A-Current Density	8.33	1	8.33	349.31	<0.0001	
B-Time	19.91	1	19.91	834.44	<0.0001	
C-NaCl	9.62 × 10^−3^	1	9.62 × 10^−3^	0.4	0.5456	
AB	0.58	1	0.58	24.18	0.0017	
AC	0.2	1	0.2	8.46	0.0227	
BC	8.05×10^−3^	1	8.05 × 10^−3^	0.34	0.5795	
A^2^	0.71	1	0.71	29.72	0.001	
B^2^	3.91	1	3.91	163.82	<0.0001	
C^2^	2.92 × 10^−4^	1	2.92 × 10^−4^	0.012	0.915	
Residual	0.17	7	0.024			
Lack of Fit	0.079	3	0.026	1.19	0.42	not significant
Pure Error	0.088	4	0.022			
Cor Total	34.03	16				

R^2^ = 0.9951, Adj R^2^ = 0.9888, Pred R^2^ = 0.9589, C.V. = 43.863%.

**Table 4 ijerph-20-01923-t004:** ANOVA analysis (Y2).

Source	Sum of Squares	df	Mean Square	F Value	*p*-Value Prob > F	
Model	2.83	9	0.31	31.58	<0.0001	significant
A-Current Density	0.061	1	0.061	6.17	0.042	
B-Time	4.65 × 10^−3^	1	4.65 × 10^−3^	0.47	0.5161	
C-NaCl	1.61	1	1.61	161.8	<0.0001	
AB	0.024	1	0.024	2.43	0.1632	
AC	0.019	1	0.019	1.89	0.2115	
BC	0.013	1	0.013	1.27	0.2974	
A^2^	0.099	1	0.099	9.95	0.0161	
B^2^	0.028	1	0.028	2.81	0.1378	
C^2^	0.91	1	0.91	91.11	<0.0001	
Residual	0.07	7	9.95 × 10^−3^			
Lack of Fit	0.055	3	0.018	5.17	0.0731	not significant
Pure Error	0.014	4	3.57 × 10^−3^			
Cor Total	2.9	16				

R^2^ = 0.976, Adj R^2^ = 0.9451, Pred R^2^ = 0.6865, C.V. = 14.182%.

## Data Availability

Data is unavailable due to privacy.

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
