# Peer review of "Electrochemical Mechanisms and Optimization System of Nitrate Removal from Groundwater by Polymetallic Nanoelectrodes"

_ijerph, 2023, doi:10.3390/ijerph20031923_

Round 1

Reviewer 1 Report

 This manuscript addressed an important issue in nitrate removal in the groundwater. To tackle this issue, the authors examined the electrochemical properties of four different electrode materials (Ti, TiO2, Cu-TiO2, and Cu-Zn-TiO2) using electrochemical measurements such as cyclic voltammetry and chronoamperometry. The Cu-Zn-TiO2-based polymetallic electrode demonstrates the best removal efficiency of nitrates from real groundwater.

 This manuscript appears to be reporting some significant results regarding the nitrate reduction made on the electroanalytical method, however, the impact is lost by the limited development of the problem and seemingly short discussion of the findings. Typically, the impact of Zn and Cu elements in the fabricated electrode materials upon the nitrate reduction reaction is not adequately explained.  I think that the results are not well discussed, and the conclusions are not meaningful in the manuscript. At the present stage, I cannot recommend it for publication in the International Journal of Environmental Research and Public Health because it does not meet the basic requirements of the journal.

 The other specific comments are listed below. I hope the comments may be helpful to improve the quality of the manuscript.

1) In the section of Materials and Methods, the authors should describe the molar concentrations of the electroplating solutions. It is better to describe the preparation procedure for the electrodes (TiO2, Cu-TiO2, Zn-Cu-TiO2). In addition, the maker and type of the electrochemical workstation should be described.

2) At line 114, the unit of ultrapure water might be incorrect.

3) In Figure 1, the authors should add the error bar to ensure the accuracy of the data analysis. In addition, it is better to add the conditions for the electrolysis.

4) In Figure 2, it is difficult to distinguish the elements (Cu, Ti, and Zn). To make it clear, the authors show the schematic depictions for the electrode structure of Cu-Zn-TiO2-based electrodes.

5) In Figure 4, Cyclic voltammetry can demonstrate the electrochemical properties of the reaction whether due to reversible- or irreversible reactions. The authors should address the characteristics of the number of cycles and the potential sweep rate dependence. I think this data is helpful to gain a deeper understanding of the electrode characteristics.

6) In figure 5, if possible, the author should analyze the diffusion-limiting process of the nitrate reduction reaction.

7) At line 301, it should be better to add the full name of ANOVA (analysis of variance?) before the abbreviation.

8) At line 379, the author reports that the removal rate in water is 100.0% after 180 minutes of electrolysis. It is better to check the accuracy of the efficiency (for instance, 99.9 ±0.05% and so on).

9) English of the manuscript has to be checked by a person familiar with electro(analytical)chemistry.

Reviewer 2 Report

In this manuscript, the authors studied Ti, TiO2, Cu-TiO2 and Zn-Cu-TiO2 electrodes as the cathode in a electrochemical system for removing nitrate from ground water. The Zn-Cu-TiO2 electrode shows the most promising performance. The reason of the best performance from Zn-Cu-TiO2 electrode is further characterized by SEM/EDS images, Tafel curves, Cyclic voltammetry and Chronoamperometry, which show that the Zn-Cu-TiO2 electrode is the most resistive to corrosion, has the highest electrochemical activity and largest electrochemical active surface area. Overall, the manuscript is good. But the English writing needs to be improved and the reviewer would suggest publication after the following issues solved.

Section 2.1, could the authors provide an figure or draw some illustrations of the setup.

Section 3.1, the removal efficiency should be clearly defined. Should also unify the name to either removal efficiency or removal rate.

Should unify the name to be either Zn-Cu-TiO2 or Cu-Zm-TiO2 or copper-zinc-titanium through the manuscript.

Line 149, typo ‘ the The’.

Line 174, should it be 9.629x10^-7 A?

Figure 4, could the authors add some indications like arrows pointing to the reduction peak locations in the CV curve for better visualization?

Line 268-269, please provide more information of the calculation of these numbers.

Eq.1-2 should be put in the correct format.

Section 3.4.3, please briefly introduce the ANOVA analysis.

How are the hope degrees defined?  The numbers in the figure captions are wrong for the last three figures. And a better explanation of the last three figures is needed. The experiments were conducted either in solution prepared with DI water or real ground water but with added nitrate. In real ground water, the nitrate concentration would be lower than the experiments conditions in this manuscript, how do the authors consider this gap between the experiments and real conditions. Another concern is that will the electrodes corrode quickly in real ground water under electrochemical conditions?

Reviewer 3 Report

The same comments are listed in the word document attached.

The authors created various electrodes for the electrochemical removal of nitrates. The nitrate removal efficiencies, corrosion rates, reactive parameters, and the active surface areas of the electrodes were determined by electrochemical methods. Electrode morphology and composition were determined by SEM and EDS techniques. Then, the authors optimized the working conditions of the electrodes with respect to removal rate, reaction time and by-product reduction. Finally, the module was tested with real wastewater with added nitrate. Their results showed the Zn-Cu-TiO2 electrode performed the best among the tested electrodes with 99% removal rate with wastewater samples. This work brings new knowledge to nitrate reduction research for environmental remediation. I believe the overall work is sound but needs major improvement in its clarity.  I believe this work is suitable for publication after the following major and minor concerns are addressed. 

Major Comments:

1.     Page 2, S2.1. The author should provide more details on the electroplating experiments such as time and voltages for the electrode fabrication. 

2.     Page 3, lines 96-105. The author should move this section to other parts of the manuscript, such as the introduction. Since this session is for describing how the experiments were conducted, the possible configurations of the potentiostat are irrelevant to the session.

3.     In the Material and Methods session. The author should also include analytical information such as how the concentrations of nitrate, nitrite, and ammonia were determined.

4.     Page 11, equations (1) and (2). The author should explain why the square root of the Y variable is used for the model, instead of the normal Y variable which is more common in Box-Behnken Design.

Minor Comments:

1.     Pages 8-9, Figure 4. Please also explain curve A (blank) and curve B (with nitrate) in the legends. Also, the resolution of (b), (c) ,and (d) figures are lower than that of figure (a). And the font used in (b) and (c) is different from (a) and (d).

2.     Page 10, line 246. Please consider changing the “delta q” to Dq or DQ. Also, the “delta bit” expression is confusing, please consider using the charge differences.

3.     Page 10, line 247. Please put a space before “Thus”.

4.     Page 10, line 251. Please consider using subscripts on “CN” and “Cd”.

5.     Page 10, Figure 5. The resolution is low.

6.     Page 11, line 285. “p-values..”, there are two periods.

7.     Page 11, lines 293-298. Please fix the formats for equations (1) and (2). There should be a space before and after operators, such as “0.22365+0.15546A”. There should be no space between multiplexed variables, such as “0.044918A C”. The last part of equation (2) is in a larger font than the rest of the equation.

8.     Page 13, lines 342-343. Please change “hopefulness” to expectation, as “hopefulness” is not the term used in mathematics.

9.     Pages 13-15. There are 3 “Figure 6”. And the resolutions are low for the first and third Figure 6.

10.  Page 14, Figure 5 legend. Please rephrase “optimization target hope degrees” to “Optimization of target variables”, to be more scientifically strict.

11.  Page 15, the third Figure 6. Please change the expression “hope” to expectations.

12.  The author should double-check the subscript and superscript formats. For example, “TiO2” were used in most of the manuscript. While “TiO2” were also used in many places such as line 74, 81-91, 125-126 etc. On line 124, the “mA/cm2” is not superscripted. On line 255, “NaNO3” and ”Na2SO4” were not subscripted.

Round 2

Reviewer 1 Report

This manuscript was revised well.

Author Response

Thank you very much for your affirmation.

Reviewer 3 Report

Thanks to the authors for their fast response. The revised manuscript is suitable for publication with only minor formatting corrections.

1. Page 2, lines 86, 87, 90, 120, and 121. There are extra spaces in "Zn -Cu-TiO2".

2. Page 3, line 109. The centimeter unit should be in lower cases as "cm".

3. Page 4, Figure 3. Please align figure 3a and 3b.

4. Page 5, line 154. There is no space before ".Nitrate".

5. Page 5, line 166. There is no space between "sidereaction".

6. Page 11, Figure 7a. The resolution for 7a is lower than the other figures.

Author Response

Response to Reviewer

Dear Reviewer:

Thank you very much for checking our manuscript (manuscript ID is ijerph-2129211), giving us the chance to revise the manuscript and proposing valuable comments! Please find the files for our revised manuscript submitted for publication in " International Journal of Environmental Research and Public Health ". Based on the reviewers’ comments and suggestions, we revised our manuscript seriously. The changed parts are highlighted in red in revised manuscript. The details about how we responded to reviewer’s comments and our modifications are in turn shown in the following part of Revision report (response to the reviewers’ specific comments) below. We sincerely hope this manuscript will be finally acceptable to be published on International Journal of Environmental Research and Public Health.

Thank you very much for all your help and looking forward to hearing from you soon.

Best wishes,

Fang Liu

Associate Professor, Dr

Institute of Transportation, Inner Mongolia University

24# Zhao Jun Road, Yu Quan District,

Huhhot, 100086, China

Tel: 86-15311183005

E-mail: 311982459@imu.edu.cn

Q1. Page 2, lines 86, 87, 90, 120, and 121. There are extra spaces in "Zn -Cu-TiO2".

Reply: Thank you for your detailed and constructive suggestion. We have deleted the extra spaces in "Zn -Cu-TiO2".

Q2. Page 3, line 109. The centimeter unit should be in lower cases as "cm".

Reply: Thank you for your advice. We have changed the units of centimeters to lowercase.

Q3. Page 4, Figure 3. Please align figure 3a and 3b.

Reply: Thank you for your advice. We have aligned figure 3a and 3b.

Q4. Page 5, line 154. There is no space before ".Nitrate".

Reply: Thank you for your advice. We have added the space before ".Nitrate".

Q5. Page 5, line 166. There is no space between "sidereaction".

Reply: Thank you for your advice. We have added the space between "sidereaction".

Q6. Page 11, Figure 7a. The resolution for 7a is lower than the other figures.

Reply: Thank you for your advice. We have replaced Figure 7a with a clearer figure.
